# The Incidence Rate, Microbiological Etiology, and Results of Treatments of Prosthetic Joint Infection following Total Knee Arthroplasty

**DOI:** 10.3390/jcm12185908

**Published:** 2023-09-12

**Authors:** Han-Kook Yoon, Ju-Hyung Yoo, Hyun-Cheol Oh, Joong-Won Ha, Sang-Hoon Park

**Affiliations:** National Health Insurance Service Ilsan Hospital, Goyang 10444, Republic of Korea; hangugy@nhimc.or.kr (H.-K.Y.); jhyoo@hanmail.net (J.-H.Y.); hyuncoh@nhimc.or.kr (H.-C.O.); hjwspine@nhimc.or.kr (J.-W.H.)

**Keywords:** arthroplasty, knee, microbiology, infection

## Abstract

Periprosthetic joint infection (PJI) remains among the most challenging and costly complications. PJI rates vary from 0.39% to 3.9% after total knee arthroplasty (TKA). This study aimed to identify the causative microorganisms involved and to report our experience of subsequent treatment of PJI following over 7000 TKAs performed over 19 years. A retrospective study was conducted on 4547 patients (7019 cases) from March 2000 to September 2019. The incidence rate of PJI was 0.5%. Gram-positive bacteria accounted for 88.8% (n = 16) of the 18 cases, and *S. aureus* was the most commonly isolated pathogen (n = 7, 38.8%). There were six cases of MSSA and one case of MRSA. *Streptococcus* species (n = 7, 38.8%) also showed the same pattern. The CoNS species (n = 2, 11.1%) and Gram-negative bacteria (n = 1, 5.5%) were also reported. *Candida* species were isolated from 1 patient (5.5%). Successful I&D and implant retention (DAIR procedures) was achieved at the final follow-up in 19 patients (82.6%). The incidence of causative microorganisms was different for each PJI onset type. The overall infection rate of PJI was less than 1%. Although the success rate of DAIR procedures is lower than the two-stage exchange arthroplasty in this study, it is possible to achieve acceptable success rates if DAIR procedures are carefully selected considering the virulence of the microorganism, duration since symptom onset, and early-onset infection.

## 1. Introduction

Total knee arthroplasty (TKA) is becoming more prevalent because of its benefits. Over 600,000 total knee arthroplasty procedures are performed annually in the United States, with that number estimated to increase to 3.5 million in 2030 [1]. The number of cases in Korea has steadily risen from 14,887 in 2001 to 75,434 in 2010 [2,3,4]. Although there have been developments in the surgical environment, instruments, and techniques, periprosthetic joint infection (PJI) remains one of the most challenging and costly complications. PJI rates vary from 0.39% to 3.9% after primary TKA [5,6,7,8], but the resulting economic burden in the United States increased annually from USD 320 million in 2001 to USD 566 million in 2009 [9]. There is a debate over the ideal treatment strategy for PJI, which has led to an international effort to advance surgical and non-surgical management [9,10,11].

This study aimed to identify the causative microorganisms involved and reports our experience of subsequent treatment of PJI following over 7000 TKAs performed over 19 years.

## 2. Methods

This retrospective study was conducted at National Health Insurance Service Ilsan Hospital, a tertiary medical center in Korea, with approval from the institutional review board. This hospital has 843 beds, and at least 800 cases of total knee arthroplasty surgeries are performed yearly. The present study included patients diagnosed with PJI after the primary TKA at a single center from March 2000 to September 2019. PJI was defined as an infection that met the Musculoskeletal Infection Society(MSIS) diagnostic criteria [12]. Patient charts were reviewed and summarized by at least two orthopedic surgeons.

The incidence rate of PJI and patient characteristics (age, sex, median follow-up time) were evaluated. Patients were classified as having an early-onset infection if infections occurred <3 months after the index surgery, delayed-onset infections if the infection occurred between 3 and 24 months following the index surgery, and late-onset if the infection occurred >24 months after the index surgery. The microbiological examination was evaluated. Treatments performed for patients with a PJI were grouped into (1) DAIR procedures (Debridement, Antibiotics, Implant Retention), (2) two-stage exchange arthroplasty, and (3) suppressive antibiotics use. It was a failure if the treatment results did not meet the Delphi international consensus criteria. Descriptive analyses were based on the percentages and frequencies of the categorical variables.

## 3. Results

A total of 4547 patients (7019 cases) were included in this study. Among these patients, 33 patients and 35 episodes of PJI were identified and diagnosed. Both knees were infected simultaneously in two patients, considered two separate episodes. The incidence rate of PJI was 0.5%. There were 5 (14.7%) male patients, and 29 (85.2%) were female. The median age of the patients was 69.2 years, which refers to the infected cases. The median follow-up period was 38.9 months.

Five cases of early-onset infection (median, 2.06; range, 0.7–2.9 [months]) were reported. Delayed-onset infection was found in 8 cases (median, 12.07; range, 4–22.6 [months]), while 22 cases were late-onset infections (median, 70.25; range, 25.9–147.2 [months]). The primary characteristics of patients with PJI at our hospital are shown in Table 1.

At least one microorganism was isolated from 18 cases, while no growth was found in 17 cases.

Cultured microorganisms included *Staphylococcus aureus* (*S. aureus*), meticillin-susceptible *S. aureus* (MSSA), meticillin-resistant *S. aureus* (MRSA), Coagulase-negative *staphylococcus* species (CoNS), *Streptococcus* species, *Escherichia coli* (*E. coli*), and *Candida* species.

Gram-positive bacteria accounted for 88.8% (n = 16) of the 18 cases, and *S. aureus* was the most commonly isolated pathogen (n = 7, 38.8%). There were six cases of MSSA and one case of MRSA. *Streptococcus* species (n = 7, 38.8%) also showed the same pattern. The CoNS species (n = 2, 11.1%) and Gram-negative bacteria (n = 1, 5.5%) were also reported. *Candida* species were isolated from 1 patient (5.5%). The details of the causative microorganisms identified in our patients are summarized in Table 2.

CoNS was the most common pathogen found in patients with early-onset PJI (40%), while MSSA was the most common pathogen isolated in patients with delayed-onset PJI (50%). *S. aureus* was not found in patients with early-onset PJI. However, *Streptococcus* species were commonly reported in patients with late-onset PJI (22.7%). *E. coli* and *candida* species were only isolated from late-onset PJI patients(4.5%, and 4.5%, respectively). Figure 1 shows the proportion of microorganisms isolated from patients with early-, delayed- and late-onset PJI.

DAIR procedures were performed in 23 cases (65.7%). The two-stage exchange arthroplasty was performed as the initial treatment in 11 cases (31.4%), involving debridement, massive irrigation, and the placement of a temporary static spacer containing vancomycin. The median interval from implant removal to reimplantation was 178 days (85–434 days). Only one patient refused the second operation; all others had successful results. The patient who refused surgery was treated with an oral antibiotic (cefdinir) and a follow-up lasting four years; however, no complications were identified.

Successful DAIR procedures were achieved at the final follow-up in 19 patients (82.6%). All six patients whose DAIR procedures were unsuccessful underwent a subsequent two-stage exchange arthroplasty and did not report reinfection. Table 3 shows the failure rates of DAIR procedures and the results of the final procedure for patients with DAIR procedures failure.

In the case of the early-onset type, DAIR procedures were performed successfully in all of the patients. However, for the six cases in which DAIR procedures were unsuccessful, the causative bacteria were MRSA in one case (100%), MSSA in two cases (50%), and *E. coli* in one case (100%). In the other two cases, no bacteria were cultured. DAIR procedures were performed successfully in cases infected by the Streptococcus species and culture negative. The mean duration of symptoms (including pain, swelling, erythema, fever, and chills) until DAIR procedures were performed, was 6.42 days for the successfully treated cases and 10 days for the cases with treatment failure. Data comparing characteristics of the DAIR procedures’ success and failure groups are presented in Table 4.

## 4. Discussion

The results of our study showed that the PJI rate for TKA is 0.5%. The incidence was similar to or slightly lower than that reported in other studies. Success rates of two-stage exchange and I&D with retention were 90.9% and 82.6%, respectively.

The most commonly cultured microorganisms are *S. aureus* (38.8%), streptococci (38.8%), CoNS (11.1%), Gram-negative bacteria (5.5%), and the Candida species (5.5%). No microorganisms were detected in this study in approximately 48.5% of infections, which is inconsistent with the results of previously published studies [8,9]. However, our microbiological data showed that Gram-positive bacteria still accounted for most of PJI. The results of our study support the use of a regimen (vancomycin in addition to piperacillin-tazobactam or cefepime [10]) that covers MRSA while simultaneously covering Gram-negative bacteria and pseudomonas when using empirical antibiotics. In this study, *S. aureus* was not found in patients with early-onset PJI. Although there was a difference in the definition of infection timing, it differed from a recent study, which reported that *S. aureus* was most commonly detected in cases of early infection [8]. It further reported that CoNS prevalence increased to 33.9% in later chronic PJI [11]. In our study, CoNS was not present in late-onset PJI; instead, Streptococcus species had a higher prevalence (22.7%), followed by *S. aureus* (13.5%). The microbiologic etiology of PJI is yet to be clarified, and each study shows slightly different results; therefore, more data will need to be accumulated.

Current guidelines for PJI treatment through DAIR procedures, one-stage exchange, or two-stage exchange arthroplasty suggest using intravenous antibiotics for 4–6 weeks [10]. Recently, there has also emerged a result showing that the treatment period of IV antibiotics may be shortened [12,13]. A two-stage exchange arthroplasty with removing all implants followed by an adequate duration of intravenous antibiotics before reimplantation has remained the gold standard treatment for PJI for over two decades [14,15]. In the case of a two-stage exchange arthroplasty, a systematic review showed similar success rates of 82–100% in eradicating infection [10]. In this study, the two-stage exchange arthroplasty had a success rate of 90.9%, while the success rate was 82.6% in patients treated with DAIR procedures. DAIR procedures were reported to have a high treatment failure rate in PJI, ranging from 34% to 84% in previous studies [16,17].

Although it has been noted that several factors affect the failure rate of PJI treatment, the type of pathogen is one of the most important factors [16]. Analyzing the episodes of unsuccessful DAIR procedures treatment in this study showed that the causative microorganisms were MRSA in one case (100%), MSSA in two cases (50%), and *E. coli* in one case (100%). According to Aggarwal et al., MRSA infection was identified as a negative predictive factor for the success of DAIR procedures [18]. In the case of our *E. coli* infection, extended-spectrum beta-lactamase positivity was noted. The higher the virulence of the microorganism, the worse the outcome of DAIR procedures.

Isolating the microorganism is important for PJI treatment success, but many culture-negative PJI continue to vex clinicians and patients [19]. Aggarwal et al. [18] found that culture-negative PJIs accounted for 15.8% of US cases and 16.1% of European cases. Using current standards for defining PJI, culture-negative cases can exceed 30% [19].

The period between the onset of symptoms and surgical intervention is also an important factor influencing the outcome [20]. Other studies have shown that high DAIR procedure success rates are associated with the duration of symptoms not exceeding one week [20,21]. There were reports like infection control was greater if DAIR procedures were performed within five days after symptom onset [20]. The difference in duration varies from study to study; the shorter the symptom onset duration, the better the result. Our findings supported this; the time until treatment after the onset of symptoms in the success group did not exceed one week. Furthermore, our study showed positive treatment results when DAIR procedures were performed in the group with early onset of infection. Boyle et al. demonstrated that between 2005 and 2014, in the United States, within 90 days after index arthroplasty, the utilization of DAIR procedures in PJI management increased significantly, with an annual increase of 3.4% (*p* < 0.001) [22]. As reported by Boyle et al. [22], the failure rate was 11.7% for patients receiving DAIR within 90 days and 31.4% for patients receiving DAIR after 90 days (*p* = 0.016). Based on these results, it is expected that the success rate of I&D would be higher in cases of PJI within 90 days after index arthroplasty and if the duration of symptoms does not exceed one week.

The National Health Insurance Service Ilsan Hospital is a large tertiary medical center in South Korea. Since 2010, we have performed more than 800 TKAs every year. The strengths of this study are the relatively long follow-up periods as a single center study and the fact that one orthopedic surgeon treated all of the patients; therefore, the treatment protocol in terms of the extent of debridement, amount of irrigation, and exchange arthroplasty technique was almost constant. We presented microbiological results for PJI and the success rate of each treatment type, which provides physicians with reliable information for clinical practice.

Our study had several limitations. First, this study had a retrospective, descriptive, and single-center design. So, this study limits the generalizability of the results. Another limitation was the relatively small sample sizes. An adequately designed prospective, randomized controlled trial is necessary to eliminate the abovementioned limitations. However, considering the incidence of PJI, this would be difficult.

## 5. Conclusions

Our study showed a difference in the incidence of causative microorganisms for each PJI onset type. The overall infection rate of PJI was less than 1%. The literature shows variations in reported failure rates of DAIR procedures. Although the success rate of DAIR procedures is lower than a two-stage exchange arthroplasty in this study, it is possible to achieve acceptable success rates if DAIR procedures are carefully selected considering the virulence of the microorganism, duration since symptom onset, and early-onset infection.

## Figures and Tables

**Figure 1 jcm-12-05908-f001:**
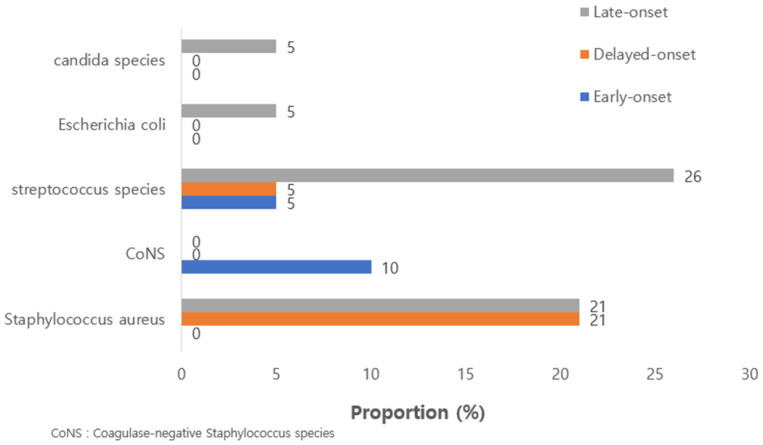
Proportion of microorganisms isolated from patients with early-, delayed- and late-onset. total knee arthroplasty prosthetic joint infection.

**Table 1 jcm-12-05908-t001:** Characteristics of 37 cases of total knee arthroplasty infection.

Characteristics	
Demographics	
No. of patients	35
No. of cases	37
Median age (range), years	69.2 (55–82)
Male	5 (14.7%)
Female	29 (85.2%)
Median follow-up (range), months	38.9 (1.8–145)
Infection	
First infection	35 (94.5%)
Re-infection	2 (5.4%)
Type of infection	
Early (≤3 months)	5 (13.5%)
Delayed (>3–24 months)	8 (21.6%)
Late (>24 months)	24 (64.8%)

**Table 2 jcm-12-05908-t002:** Microorganisms isolated from cases with post-total knee arthroplasty prosthetic joint infection.

Microorganism	No. (%)
Gram-positive bacteria	
*Staphylococcus aureus*	8 (42.1)
Methicillin-susceptible	6 (31.5)
Methicillin-resistant	2 (10.5)
Coagulase-negative *staphylococcus* species	2 (10.5)
*Streptococcus* species	7 (36.8)
Gram-negative bacteria	
*Escherichia coli*	1 (5.2)
*Candida* species	1 (5.2)
No growth	20

**Table 3 jcm-12-05908-t003:** Frequency of final procedure in the initial group of failed irrigation and debridements.

Type of Treatment	No. (%)
DAIR procedures	20 (54)
Two-stage exchange arthroplasty	16 (43.2)
Antibiotics	1 (2.7)

**Table 4 jcm-12-05908-t004:** Frequency of final procedure in the initial group of failed irrigation and debridements.

Procedure	No. (%)	Type of Isolated Organism	Time from Symptoms to Surgery
Total I&D	25	Streptococcus species (15), no growth (10)	6.42 days
Failure of initial I&D	6 (24)	MRSA (1), MSSA (2), *E. coli* (1), no growth (2)	10 days
Two-stage exchange arthroplasty	5 (20)		
Arthrodesis	1 (4)		

## Data Availability

The datasets used and analysed during the current study available from the corresponding author on reasonable request.

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
