# Peer review of "The Incidence Rate, Microbiological Etiology, and Results of Treatments of Prosthetic Joint Infection following Total Knee Arthroplasty"

_jcm, 2023, doi:10.3390/jcm12185908_

Round 1
Reviewer 1 Report
The overall logic of the article is rigorous, the wording is accurate, and it has research significance, but there are some minor problems that need to be corrected.
Manuscript: jcm-2515365
The manuscript is entitled “The incidence rate, microbiological etiology, and results of treatments of prosthetic joint infection following total knee arthroplasty”.
In this work, the authors showed a difference in the incidence of causative microorganisms for each PJI onset type. Although the success rate of I&D is lower than a two-stage exchange in this study, it is possible to achieve acceptable success rates if I&D is carefully selected considering the virulence of the microorganism, duration since symptom onset, and early-onset infection. However, we think there are still some issues in this manuscript need to be corrected by the author.
(1) The retrospective, descriptive design of this study limits generalizability of the results..
(2) Larger prospective randomized controlled trials are recommended to increase sample diversity and reliability of results.
(3) Further research on other factors that may affect the occurrence of PJI and treatment outcomes, such as patient health, surgical technique, postoperative care, etcIn the experiment of cytotoxic activity, it is not clear which kind of cells are used.
(4) In the method, description of microbial culture: For asserted cultured microorganisms, it is recommended to provide some number or percentage of culture results to support these assertions.
The article has a high level of English and accurate wording
Author Response
The manuscript is entitled “The incidence rate, microbiological etiology, and results of treatments of prosthetic joint infection following total knee arthroplasty”.
In this work, the authors showed a difference in the incidence of causative microorganisms for each PJI onset type. Although the success rate of I&D is lower than a two-stage exchange in this study, it is possible to achieve acceptable success rates if I&D is carefully selected considering the virulence of the microorganism, duration since symptom onset, and early-onset infection. However, we think there are still some issues in this manuscript need to be corrected by the author.
- The retrospective, descriptive design of this study limits generalizability of the results..
: Thanks for your comment. We revised it.
- Larger prospective randomized controlled trials are recommended to increase sample diversity and reliability of results.
: Thanks for your comment. It was not done in this study, but we will make it possible to promote it with such a research design in future studies.
(3) Further research on other factors that may affect the occurrence of PJI and treatment outcomes, such as patient health, surgical technique, postoperative care, etcIn the experiment of cytotoxic activity, it is not clear which kind of cells are used.
: Thanks for your comment. It is difficult to accurately identify the cause of total knee arthroplasty infection because choosing one of the categories is hard. And More than 30% of cases where the causative strain cannot be identified.
(4) In the method, description of microbial culture: For asserted cultured microorganisms, it is recommended to provide some number or percentage of culture results to support these assertions.
: Thanks for your comment. I added % to the table and the result.
Reviewer 2 Report
Dear Editor,
Thank you for allowing me to review the article entitled “The incidence rate, microbiological etiology, and results of treatments of prosthetic joint infection following total knee arthroplasty.”
Abstract
Line 8, delete the word “thus”
Lines 8-10; the authors must use the same tense (past or present).
Line 13: better use past tense when reporting your findings
Introduction
Use new paragraph for the aim of the study (line 34).
Methods
Please provide some information regrading the institution. How many beds does it have and how many are for Orthopaedic patients.
Line 43: provide reference after mentioning the MIS criteria.
Lines 50-52: This belongs to the results’ section. You should mention in the Methods section that microbiological examination was evaluated.
Line 53: (I&D): In this category were the mobile parts of the arthroplasty changed?? (polyethylene). This is of utmost importance to be clarified. I would be more precise about these definitions. For example since you mention antibiotic suppression, it is not clear that the other two categories received antimicrobial treatment which I believe did. I&D has been documented in the literature as DAIR procedure (Debridement, Antibiotics, Implant Retention) better use this in your definitions. “Two-stage exchange “ should be “two-stage exchange arthroplasty” throughout the text.
Results
Line 58: You have already mentioned that in the methods. Please start with your results, eg A total of 4,547 patients were included in the study etc…
Provide the demographics of the total sample – not only the for the 33 infected cases, otherwise this is not an analysis of almost 5000 patients but just a retrospective cohort of 33 patients.
Line 61: the median age refers to the infected cases or to the total 4547 patients?
Lines 63-66: better use the word “cases” instead of episodes- do that for the whole text.
Line 83: now the authors explain that polyethylene was exchanged; this information needs also to be in the methods section where they define the 3 methods of treatment, as previously commented.
Line 93-94: “None of the patients with unsuccessful results after I&D eventually required further suppressive antibiotics treatment”: it is not clear what the authors mean. Please rephrase
Disucssion
Line 117 : Better use past tense, since you refer to the results of the concluded study.
Lines 118-119: the authors state that the relative high incidence of no-cultured PJI is in line with the literature. This is quite controversial since many studies report wide range of this clinical entity. Please elaborate more on this topic, it is quite interesting and challenging in every day clinical practice.
Lines 120-125: You refer to empirical treatment. You mean in cases that the cultures are negative? Or In general, which other PJI cases are treated with empirical treatment? Please rephrase to make it clearer to the reader.
Lines 133-134: IV antibiotics for 4-6 weeks is not mandatory; antibiotic therapy for this period yes, not necessarily iv .
Please see these references regrading the matter:
1) Barnes E, Russ-Friedman C, Ross E, Aranda E, Whitt B. The Use and Monitoring of Oral Antibiotics for Treatment of Prosthetic Joint Infections. J Surg Orthop Adv. 2021 Winter;30(4):256-262.
2) Bouji N, Wen S, Dietz MJ. Intravenous antibiotic duration in the treatment of prosthetic joint infection: systematic review and meta-analysis. J Bone Jt Infect. 2022 Sep 19;7(5):191-202. doi: 10.5194/jbji-7-191-2022.
The issue of duration and type (IV or oral) is a debatable matter in PJIs and rather interesting in discussing. I would elaborate on this matter more.
Lines 159-164: the authors now use the DAIR acronym and then I&D : are they different procedures??? What is the difference of I& D when compared to DAIR? If there is no difference, as I have understood I& D is rather misleading and should be changed to DAIR , as I have already reported in the Methods section.
I appreciate the effort of the authors.
A few revisions should be made to make the manuscript more precise and clear to the reader.
I wish the authors best of luck in the reviewing process and congratulate them for their efforts in this paper.
Minor Editing is required. I mention the parts that need rephrasing in the Reviewer's comments section.
Author Response
Reviewer 2
Abstract
Line 8, delete the word “thus”
: Thanks for your comment. I removed it.
Lines 8-10; the authors must use the same tense (past or present).
: Thanks for your comment. I revised the sentence.
Line 13: better use past tense when reporting your findings
: Thanks for your comment. I revised the sentence.
Introduction
Use new paragraph for the aim of the study (line 34).
: Thanks for your comment. I revised the sentence.
Methods
Please provide some information regrading the institution. How many beds does it have and how many are for Orthopaedic patients.
: Thanks for your comment. This hospital has 843 beds, and at least 800 cases of total knee arthroplasty surgeries are performed yearly
Line 43: provide reference after mentioning the MIS criteria.
: Thanks for your comment. I added the reference
Lines 50-52: This belongs to the results’ section. You should mention in the Methods section that microbiological examination was evaluated.
: Thanks for your comment. I revised the sentence.
Line 53: (I&D): In this category were the mobile parts of the arthroplasty changed?? (polyethylene). This is of utmost importance to be clarified. I would be more precise about these definitions. For example since you mention antibiotic suppression, it is not clear that the other two categories received antimicrobial treatment which I believe did. I&D has been documented in the literature as DAIR procedure (Debridement, Antibiotics, Implant Retention) better use this in your definitions. “Two-stage exchange “ should be “two-stage exchange arthroplasty” throughout the text.
: Thanks for your comment. I revised the it. As you recommended to me, I organized it into three categories.
Results
Line 58: You have already mentioned that in the methods. Please start with your results, eg A total of 4,547 patients were included in the study etc…
: Thanks for your comment. I revised the sentence.
Provide the demographics of the total sample – not only the for the 33 infected cases, otherwise this is not an analysis of almost 5000 patients but just a retrospective cohort of 33 patients.
Line 61: the median age refers to the infected cases or to the total 4547 patients?
: It refers to the infected cases
Lines 63-66: better use the word “cases” instead of episodes- do that for the whole text.
: Thanks for your comment. I revised the word throughout the whole text.
Line 83: now the authors explain that polyethylene was exchanged; this information needs also to be in the methods section where they define the 3 methods of treatment, as previously commented.
: Thanks for your comment. I revised the it. As you recommended to me, I organized it into three categories.
Line 93-94: “None of the patients with unsuccessful results after I&D eventually required further suppressive antibiotics treatment”: it is not clear what the authors mean. Please rephrase
: It means that all patients got better after the DAIR procedure or after two-stage exchange arthroplasty
Disucssion
Line 117 : Better use past tense, since you refer to the results of the concluded study.
: Thanks for your comment. I revised the sentence.
Lines 118-119: the authors state that the relative high incidence of no-cultured PJI is in line with the literature. This is quite controversial since many studies report wide range of this clinical entity. Please elaborate more on this topic, it is quite interesting and challenging in every day clinical practice.
: Thanks for your comment. We added the phrase like this. Isolating the offending microorganism is paramount for PJI treatment success However, a significant number of culture-negative PJI continue to vex clinicians and patients. Aggarwal, et al. found that culture-negative PJIsaccounted for 15.8% of US cases and 16.1% of European cases, and a high percentage of culture-negative PJIs have been reported in other published series. For example, in a retrospective review of 785 PJIs treated with 2-stage exchange arthroplasty over a 14-year period at a single institution, Bjerke-Kroll, et al. found that 21.3% of cases were culture-negative. Using current standards for defining PJI, culture-negative cases can exceed 30%.
Lines 120-125: You refer to empirical treatment. You mean in cases that the cultures are negative? Or In general, which other PJI cases are treated with empirical treatment? Please rephrase to make it clearer to the reader.
: Thanks for your comment. It means that we used antibiotics empirically before the strain is cultured.
Lines 133-134: IV antibiotics for 4-6 weeks is not mandatory; antibiotic therapy for this period yes, not necessarily iv .
Please see these references regrading the matter:
1) Barnes E, Russ-Friedman C, Ross E, Aranda E, Whitt B. The Use and Monitoring of Oral Antibiotics for Treatment of Prosthetic Joint Infections. J Surg Orthop Adv. 2021 Winter;30(4):256-262.
2) Bouji N, Wen S, Dietz MJ. Intravenous antibiotic duration in the treatment of prosthetic joint infection: systematic review and meta-analysis. J Bone Jt Infect. 2022 Sep 19;7(5):191-202. doi: 10.5194/jbji-7-191-2022.
The issue of duration and type (IV or oral) is a debatable matter in PJIs and rather interesting in discussing. I would elaborate on this matter more.
: Thanks for your comment. I added the references. However, research on the duration of antibiotic use seems to be beside the point. In the past, it was a principle to write long, but recently, it is thought that there is no problem even if it is short.
Lines 159-164: the authors now use the DAIR acronym and then I&D : are they different procedures??? What is the difference of I& D when compared to DAIR? If there is no difference, as I have understood I& D is rather misleading and should be changed to DAIR , as I have already reported in the Methods section.
: Thanks for your comment. I revised the it. As you recommended to me, I organized it into three categories.
Reviewer 3 Report
Good retrospective study on incidence , microbiology etiology and results of prosthetic joint infection (PJI)fvollowing total knee arthroplasty from a large single tertiary centre in Island Korea over 19 years from 2000-2019.
Introduction was sufficient to introduce extent of PJI. Methods explained common microbiology ethology and timeline of PJI.
Results were drawn from 4547 patients from 1 centre (Ilsan, South Korea). If a national joint registry was available (like in Australia) observational analysis could be drawn from larger numbers. Tables detailed defined dempographics and health characteristics of PJI population.
Discussion explains ethology of common PJI microbiology and success of treatment of PJI with irrigation and debridement (I& D).
Conxclusions drawn were appropriate from evidence and results tables
Appropriate
Author Response
Thank you very much for taking a good view of my paper. Currently, we are not able to freely use Korean big data, so we conducted a single center study.